# The Role of circRNAs in Human Papillomavirus (HPV)-Associated Cancers

**DOI:** 10.3390/cancers13051173

**Published:** 2021-03-09

**Authors:** Patrizia Bonelli, Antonella Borrelli, Franca Maria Tuccillo, Franco Maria Buonaguro, Maria Lina Tornesello

**Affiliations:** 1Molecular Biology and Viral Oncology, Istituto Nazionale Tumori—IRCCS—Fondazione G. Pascale, 80131 Napoli, Italy; f.tuccillo@istitutotumori.na.it (F.M.T.); f.buonaguro@istitutotumori.na.it (F.M.B.); m.tornesello@istitutotumori.na.it (M.L.T.); 2Innovative Immunological Models, Istituto Nazionale Tumori—IRCCS—Fondazione G. Pascale, 80131 Napoli, Italy; a.borrelli@istitutotumori.na.it

**Keywords:** circRNAs, HPV-associated cancers, squamous cell carcinoma, biomarkers

## Abstract

**Simple Summary:**

Circular RNAs (circRNAs), a new class of non-coding RNAs, are aberrantly expressed in several cancer types. It has been shown that circRNAs are involved in tumorigenesis and cancer progression, as well as in drug resistance. Some circRNAs are useful markers of diagnosis and prognosis. In this review, we examined the role of circRNAs in HPV-associated cancers, highlighting their importance as biomarkers in the diagnosis, prognosis, and therapy of anogenital and oropharyngeal and oral cancers.

**Abstract:**

Circular RNAs (circRNAs) are a new class of “non-coding RNAs” that originate from non-sequential back-splicing of exons and/or introns of precursor messenger RNAs (pre-mRNAs). These molecules are generally produced at low levels in a cell-type-specific manner in mammalian tissues, but due to their circular conformation they are unaffected by the cell mRNA decay machinery. circRNAs can sponge multiple microRNAs or RNA-binding proteins and play a crucial role in the regulation of gene expression and protein translation. Many circRNAs have been shown to be aberrantly expressed in several cancer types, and to sustain specific oncogenic processes. Particularly, in virus-associated malignancies such as human papillomavirus (HPV)-associated anogenital carcinoma and oropharyngeal and oral cancers, circRNAs have been shown to be involved in tumorigenesis and cancer progression, as well as in drug resistance, and some are useful diagnostic and prognostic markers. HPV-derived circRNAs, encompassing the HPV E7 oncogene, have been shown to be expressed and to serve as transcript for synthesis of the E7 oncoprotein, thus reinforcing the virus oncogenic activity in HPV-associated cancers. In this review, we summarize research advances in the biogenesis of cell and viral circRNAs, their features and functions in the pathophysiology of HPV-associated tumors, and their importance as diagnostic, prognostic, and therapeutic targets in anogenital and oropharyngeal and oral cancers.

## 1. Introduction

Circular forms of RNA (circRNA) were first identified by electron microscopy in the cytoplasm of several eukaryotic cells about 40 years ago [1]. Later, Cocquerelle et al. identified ETS proto-oncogene 1, transcription factor (*ETS1*) transcripts in the form of stable circular RNAs, containing only exons in the correct genomic order, and considered such structures as RNA splicing defects [2]. Only recently, it has been shown that thousands of genes produce highly conserved circRNAs that regulate numerous biological processes [3,4,5]. circRNAs have been shown to originate from newly identified alternative regulated splicing, called back-splicing, wherein a downstream donor site is linked to an upstream acceptor site to form a circular molecule [6]. In the past few years, circRNAs have remained mostly unrecognized due to the limitations of quantitative reverse transcription PCR (RT-qPCR) techniques, which are generally performed with primers that are unable to distinguish linear from circular RNAs due to being designed from linear genome sequences rather than being “outward facing” for evidencing back-spliced RNA species; there are also limitations of next-generation sequencing (NGS) data analyses, which discard the spanning back-splicing junction reads that do not map to the linear reference genome.

Furthermore, circRNAs, having no poly-adenylic acid (poly-A) tails, escape in the mRNA purification phase from the rRNA used to prepare sequencing libraries [7,8]. The use of primers designed to span circRNA back-spliced sequences, the improvement of sequencing technologies and mapping algorithms, and the use of ribosomal RNA depletion strategies, which enable the sequencing of non-polyadenylated RNAs, have allowed genome-wide studies of circRNAs [3,9,10,11,12]. circRNAs are very stable because they have no free ends, so they cannot be attacked by exonucleases [3,13], but they may contain sites of endonuclease attack. A recent study has shown that the N^6^-methylation of RNA adenosine facilitates recognition by endonucleases capable of degrading circRNAs [14]. Despite this, circRNAs appear to be more stable than linear RNAs [15]. circRNAs are deregulated in several human tumors, including tumors caused by infectious agents such as oncoviruses. They play an essential role in cancer pathogenesis by influencing many cancer characteristic signs [16,17]. circRNAs have great potential as biological markers in tumors; specific circRNAs are often expressed in one particular disease, and are predictive of outcome [18,19]. They are also detectable in biological fluids, blood, and saliva, a fundamental element for non-invasive biomarkers [20,21,22,23].

This review examined the current state of knowledge of circRNAs with a particular focus on HPV-associated cancers such as anogenital cancers, including carcinoma of the cervix, vulva, vagina, penis, and anus, as well as oropharyngeal and oral cancers, whose frequency is increasing worldwide. Moreover, for the latter cancers, the presence of HPV is essential for disease management, and the identification of new and effective biomarkers of viral oncogenic activity—such as circRNAs—is very relevant for patient stratification and treatment, as they already are for other neoplastic pathologies.

## 2. Classification, Biogenesis and Degradation of circRNAs

circRNAs are synthesized by pre-mRNA transcripts of protein-coding genes, but unlike linear mRNAs, they are circularized by covalent binding of the 3′ and 5′ ends by a single back-splicing [24]. The nomenclature of circRNAs follows parental genes or specific functions, although there is currently no standard nomenclature system for circRNAs [25]. Indeed, several circRNAs databases use different naming systems, which can lead to mistakes [26]. circRNAs are classified into three categories: *ecircRNA* (exonic circRNA), which contains only exons, *ciRNA* (intronic circRNA) that originate from introns, and *EIciRNA* (exon-intron circRNA) formed by exons and introns. Jeck et al. proposed two circularization models: (1) the lariat guide, and (2) the intronic coupling guide [27].

The first model is associated with the exon skip. A connection is established between the 3′ end of the donor exon with the 5′ end of the acceptor exon to form a lariat (loop) structure containing exons. The spliceosome joins the lariat, and a circle of exons is obtained after removing the introns. The other model is based on the direct base pairing of complementary sequence motifs in the transcripts. The ecircRNA and EIciRNA are derived from intron elimination or intron retention, respectively. Flanking Alu elements, as well as other repeated inverse sequences, play a significant role in the biogenesis of circRNAs [28]. A recent study identified a subset of mammalian circRNAs derived from mammalian-wide interspersed repeat (MIR) mediated back splicing [29].

The intron-guided circularization pattern occurs more frequently than the lariat model [3]. A new mechanism has been proposed for the genesis of circRNAs, linked to failure in debranching [30]. Usually, the intronic lariat is debranched and subsequently degraded. A debranching failure may occur and lead to the formation of ciRNAs. Recent studies have suggested that circRNAs can be generated by the union of RNA with RNA-binding proteins (RBP). Muscleblind protein (MBL) or quaking protein (QKI) can bind to flanking introns and mediate circularization [31,32]. Furthermore, through the tRNA splicing endonuclease complex, pre-tRNAs, in the presence of a conserved tRNA sequence motif, can give rise to tRNA intronic circular RNAs (tricRNAs) [33]. Figure 1 summarizes the classification and biogenesis of circRNAs.

Eukaryotic circRNAs can also originate from the mechanisms of trans-splicing or exon scrambling. In the trans-splicing mechanism, exons of two different RNAs join, then the rearrangement of exons in a circular orientation occurs with back-splicing mediation [3]. Scrambled exons can occur due to genomic rearrangements, tandem duplications, trans-splicing, or back-splicing [34]. Cancer-associated chromosomal translocations lead to the production of fusion circRNAs that contribute to cell transformation by influencing cell viability and resistance to therapy, and promoting tumor in vivo models [35].

Not much is known about the degradation mechanisms of circRNAs. Park et al. reported that linear and circular RNAs containing N^6^-methyladenosine are degraded by an endoribonucleolytic YTH N^6^-methyladenosine RNA binding protein 2 (YTHDF2)- heat-responsive protein 12 (HRSP12)—ribonuclease P and MRP (RPP-MRP) cleavage pathway. [14]. On the other hand, RNase latent (RNase L) is responsible for the degradation of circRNAs after viral infection [36]. Fischer et al. have observed that in tumor cells, the degradation of circRNAs is regulated by two RBPs—UPF1 RNA helicase and ATPase (UPF1)—and G3BP stress granule assembly factor 1 (G3BP1) [37]. Cancer cells can also remove cytoplasmic circRNAs through microvesicles and exosomes [38,39].

## 3. Functions of circRNAs

The class of circRNAs are a new type of molecule to be explored, as not much is known about their role. The functions of circRNAs, highlighted in the studies carried out to date include the regulation of RNA transcription, miRNA and protein sponging, and translation into small proteins [40].

circRNAs are found in the nucleus and the cytoplasm of cells, although the transport mechanisms are unknown, and they may regulate gene transcription [6]. The mechanisms by which circRNAs regulate gene transcription are varied, and cellular localization of circRNAs could indicate the potential function [41]. circRNAs that are localized in the nucleus perform the function of gene regulation at the transcription level. Both EIciRNAs and ciRNAs interact with specific molecules located in the nucleus: EIciRNAs adopt a regulatory strategy for transcriptional control through specific RNA-RNA interactions [42]; EIciRNAs, such as circEIF3J and circPAIP2, can bind to U1 small nuclear ribonucleoprotein A (U1SnRNP), recruiting polymerase II (POLR2) to increase their parental gene expression [42]. Ci-ankrd52 and ci-sirt7 also regulate gene transcription by association with POLR2 machinery, acting as positive regulators of POLR2 transcription [30]. Furthermore, some circRNAs can act on their pre-mRNA to produce circular transcripts through alternative splicing. In this way, they regulate gene expression, inhibiting the production of canonical proteins [43].

Several circRNAs are exported into the cytoplasm, where they interact with and sequester RBPs and miRNAs. The circRNA and RBP interactions have a double effect, influencing protein expression and function and regulating the synthesis and degradation of circRNAs. RBPs play a role in various cellular processes, such as proliferation, differentiation, apoptosis, senescence, and response to oxidative stress, as they regulate RNA at the post-transcriptional level [44,45,46]. Stable RNA/protein complexes consisting of circRNAs and proteins have been identified, including Argonaute (AGO) [24,47], MBL [48], QKI [32], and eukaryotic translation initiation factor 4A3 (EIF4A3) [49]. circRNAs, acting as protein sponges, can regulate RBPs, modulating the mRNA translation process [50]. Most circRNAs have been predicted to interact with RBPs through specific binding sites, even though bioinformatics analyses have predicted very few enrichments in RBP binding sites. Recent studies have shown that RNA-RBP interactions are influenced by the RNA molecule tertiary structure [51]. The RBPs regulate the circRNAs circularization. Among the RBPs that have this feature are included QKI, FUS RNA binding protein (FUS), and MBNL1. Ashwal-Fluss et al. identified, for the first time, the involvement of MBL in the circularization of exons [31]. The binding of QKI to the intronic regions close to the exons to be circularized could facilitate RNA looping and back-splicing [32]. The effects that an RBP can have on regulating the formation of a circRNA depend on the type of circRNA, cell, and tissue. Recently, RNA-binding protein 3 (RBP3) has been identified as a potential player in regulating the production of circRNAs and, particularly, of SCD-circRNA 2, modulating cell proliferation in human hepatocellular carcinoma [52].

RBPs can have both an activator and an inhibitor function in the formation of circRNAs, regulating their expression through multiple mechanisms [16,53].

Additionally, circRNAs can form a functional complex with proteins. Usually, cyclin-dependent kinase 2 (CDK2) interacts with cyclin A (CCNA) and cyclin E (CCNE) for cell cycle progression, while cyclin-dependent kinase inhibitor 1A (CDKN1A) inhibits these interactions by blocking cell cycle progression. Circ-Foxo3 has been shown to delay cell cycle progression by forming a ternary complex with CDKN1A and CDK2 [54]. The overexpression of circ-Foxo3 results in a more significant interaction between CDK2 and CDKN1A, while siRNA-mediated knockdown reduces the protein-protein interaction. This study demonstrated how circRNA could act as a scaffold in regulating protein-protein interactions. Some circRNAs can modify the stability of mRNAs. A circular antisense RNA molecule has been shown to stabilize CDR1 mRNA [55]. Mouse circRasGEF1B, inducible with lipopolysaccharide (LPS), facilitates the stabilization of mature intercellular adhesion molecule 1 (*ICAM*-*1*) transcripts in macrophages [56].

Although circRNAs are classified as non-coding RNAs, recent studies have shown that some circRNAs can code for proteins [40]. Indeed, several circRNAs have open-reading frames and can be translated into peptides or proteins. Among these, circ-FBXW7, circ-ZNF609, and circMbl have been reported to be translated into proteins. Cir-FBXW7 encodes a protein of 185 amino acids that plays an active role in repressing glioma tumorigenesis [57]. Circ-ZNF609 regulates myogenesis through protein expression. Circ-ZNF609 contains both a start and stop codon, and an internal ribosome entry site in its untranslated region, allowing for translation in a cap-independent manner [58]. CircMbl encodes a protein detected by mass spectrometry on fly head extracts [59]. Studying the translation of circRNAs could reveal a hidden human proteome and improve understanding of the importance of circRNA in human cancers.

The most common function of circRNAs is associated with miRNA sponging, which is defined as a partial antisense interaction between a non-coding RNA and an miRNA [60]. The circRNAs, acting as sponges of the miRNAs, block the miRNA binding to their target mRNAs, thus regulating their expression. The circRNAs, in their sequences, possess miRNA response elements (MREs) that bind miRNAs, limiting their RNA regulation functions [47,61]. circRNAs have many binding sites, even more than miRNAs [62,63]. mRNAs and circRNAs compete by binding the same miRNA via MREs [64]. Mathematical models have indicated that sponging of miRNAs depends on the mobility mechanism of miRNAs, described as an intermittent active transport mechanism, an alternation between active transport and Brownian motions, suggesting that the interaction between miRNAs and their targets depends on the concentration, affinity, and localization of the molecules [65]. ciRS-7/CDR1as possess more than 70 miRNA interaction sites [47].

However, some studies have questioned the role of circRNAs as sponges, because many circRNAs do not have or have a limited number of miRNA binding sites [7,66]. Furthermore, tissue circRNAs are so limited that they could not regulate RNA transcription [67]. It has been demonstrated that, by using a bioinformatic algorithm, only two human circRNAs have more predicted binding sites than expected by chance [68].

## 4. circRNAs as Biomarkers for Diseases

circRNAs, with their multiple functions, are involved in many biological and pathological processes; they influence the progression of diseases, including viral infections and cancer [69]. In cancer, circRNAs influence the malignant phenotype by regulating specific cancer-related pathways and exert both an anti-cancer and pro-cancer action by acting as a tumor suppressor or oncogene [70,71]. They can be used as therapeutic targets in treatment. circRNAs are insensitive to the action of ribonucleases [72], and their levels of expression are lower than those of mRNAs [27,54,65]. However, circRNAs act in cells and tissues in a type-specific manner and are stably expressed in saliva, blood, tissues, and exosomes [73,74,75]. For all these characteristics, ribonucleases resistance, tissue and type-specific function, and the presence in biological fluids, the circRNAs could be biomarkers in the diagnosis and prognosis of various diseases [76,77].

## 5. circRNAs in Human Papillomavirus (HPV)-Associated Cancers

circRNAs play essential roles in many physiological and pathological processes, such as the cell cycle, apoptosis, vascularization, tumorigenesis, invasion, and metastasis [78]. circRNAs play a role in the carcinogenesis and development of HPV-associated cancers [79]. The continued study of circRNA and tumor interactions is required to identify new circRNAs as molecular markers or potential therapeutic targets, which could be promising for early diagnosis, cancer prognosis, and even cancer therapy applications.

## 6. Human Papillomaviruses

Alpha human papillomaviruses (HPVs) are the most common sexually transmitted infectious agents [80,81]. The recognition of the oncogenic role of HPV infection in the pathogenesis of about five percent of human tumors has stimulated many epidemiologic and molecular studies and the development of vaccines to reduce HPV-induced malignancies [82]. The group of mucosal “oncogenic” HPVs, particularly genotype HPV16, may infect the anogenital tract and the head and neck region and be associated with almost all cervical and anal carcinomas and with a variable proportion of other low tract anogenital tumors, including vaginal, vulvar, and penile cancers [83], as well as with cancers of the head and neck region, such as oropharyngeal and oral carcinomas (Figure 2).

Interactions between the oncoproteins of the high-risk HPVs, E6 and E7, and the proteins of the anti-oncogenes tumor protein P53 (*TP53*) and RB transcriptional corepressor 1 (*RB1*) of the host cells, are involved in HPV-related carcinogenesis. Carcinogenesis involves the integration of the viral genome into host cell chromosomes, inhibition of tumor suppressor genes, mutations, and, ultimately, the development of dysplasia or tumors [84].

## 7. HPVs and circRNAs

Recently, virus-derived circRNAs have been described in gammaherpesviruses [85]. Zhao et al. reported that oncogenic HPVs generate circRNAs, some of which encompass the E7 oncogene (circE7) [86]. HPV16 circE7 is detectable in HPV16-transformed cells. It is N^6^-methyladenosine (m6A) modified, preferentially localized to the cytoplasm, and associated with polysomes. HPV16 circE7 is translated to produce E7 oncoprotein. In the HPV-16 positive cervical cancer cells CaSki, specific disruption of circE7 reduces E7 protein levels, inhibiting cell cancer growth, both in vitro and in tumor xenografts. CircE7 is also present in TCGA RNA-Seq data from HPV-positive cancers and cell lines with only episomal HPVs. From this, it is evident that virus-derived, protein-encoding circular RNAs are biologically functional and linked to the transforming properties of some HPVs [86]. CircE7 is required for optimal E7 expression in CaSki cells. The cytoplasmic localization of circE7 may help explain the remarkable contribution of circE7 to E7 oncoprotein expression. Elevated circE7 levels have been detected in other HPV-associated cancers. High levels of circE7 have been identified as positive prognostic markers in anal squamous cell carcinomas (ASCC) [87].

Persistent infections of the squamous or glandular epithelium by oncogenic HPVs represent the primary cause of cervical tumors. High-risk HPVs, HPV16, and HPV18 are the most frequently detected subtypes in the world, making up 70% of invasive cervical cancer cases [88]. HPV E6 and E7 proteins play a role in the transforming properties associated with HPV infection and cancer development, as they deregulate p53 and pRb signaling, respectively [89,90]. In the study conducted by Zheng et al., the E7 oncogene expression was investigated on the expression profiles of circRNAs in CaSki cells [91]. After the silencing of endogenous E7 expression, 526 deregulated circRNAs, including 352 up-regulated and 174 down-regulated circRNAs, were identified. Some of these modulated circRNAs target miRNAs reported in the pathogenesis of cervical cancer [92,93,94]. With its essential role in human cervical cancer, mTOR signaling was the pathway involved in this study [95,96]. Overall, studies have shown that patients with HPV-16-positive tumors and high circE7 expression levels exhibit better overall survival (OS) than patients with HPV-16-positive tumors but with low levels of circE7. CircE7 could be a biomarker in HPV-associated malignancies.

Little is known about the interactions between viruses and host circRNAs; however, it has been reported that in virus-infected cells, the expression patterns of circRNAs are altered compared to the control cells [97,98]. The viruses probably use circRNAs for their progression [69,86].

## 8. circRNAs and Anogenital Cancers

### 8.1. circRNAs and Cervical Carcinoma

Cervical cancer ranks as the fourth most frequently diagnosed cancer, and fourth as the leading cause of cancer death worldwide. In fact, 570,000 new cases were diagnosed, and 311,000 deaths, in 2018 [99]. Regarding the incidence and mortality, cervical cancer ranks second [99]. HPV, with its 12 oncogenic types [80], represents a necessary but not sufficient cause of cervical cancer [100]. Considerable progress has been made in recent decades, but there is still little knowledge of cervical cancer carcinogenesis mechanisms. It has recently been shown that circRNAs are differentially expressed in tumor tissues [101], making them potential tumor treatment targets. Furthermore, circRNAs can be considered new tumor biomarkers [101,102,103].

Some circRNAs identified and studied in cervical carcinoma are biomarkers for the diagnosis and progression of the disease and as probable targets for therapy; others have proved to be biomarkers for diagnosis and/or disease progression or target therapy, others for disease progression and targeted therapy, and still others only for diagnosis, disease progression or as targets for specific therapy.

CDR1as/ciRS-7 (hsa_circ_0001946) contains elements that bind miR-7. The expression of CDR1as is higher in cervical carcinoma than in para-carcinoma tissues, and its expression is the opposite of miR-7. In cervical carcinoma cells, overexpression of CDR1as enhances focal adhesion kinase (FAK) levels by suppressing the activity of miR-7. FAK overexpression promotes proliferation, invasion, and metastasis of cervical carcinoma, suggesting the close cooperation between CDR1as/miR-7/FAK [104].

The expression of circRNA8924, upregulated in cervical cancer, is associated positively with tumor size, stage, and myometrial invasion, demonstrating a role in cervical cancer progression. CircRNA8924 acts as a sponge for miR-518d-5p and influences the chromobox 8 (*CBX8*) gene expression closely associated with histogenesis, invasion and metastasis, and prognosis of cervical cancer. CircRNA8924 could be a biomarker for diagnosis and disease progression and a therapeutic target for cervical cancer [105].

CircRNA-000284 (circHIPK3) is frequently upregulated in cancer patients [106] and cervical cancer cell lines [107]. Compared with cervical epithelial cells, the expression levels of circRNA-000284 significantly increase in cervical cancer cells. CircRNA-000284 functions as a tumor oncogene by promoting cell proliferation and invasion. By computational analysis, miR-506 was associated with circRNA-000284. The direct target of miR-506 was identified as snail family transcriptional repressor 2 (*SNAI2*). CircRNA-000284 could regulate cell proliferation and invasion by sponging miR-506 and influencing the expression of *SNAI2*. CircRNA-000284 may serve as a promising therapeutic target for cervical cancer patients.

Hsa_circ_0000263 could influence cell proliferation, migration, the cell cycle, and apoptosis in cervical carcinoma cells. It has been demonstrated that hsa_circ_0000263 is significantly upregulated in cervical cancer cells [108]. The knockdown of hsa_circ_0000263 inhibits cell proliferation and migration. hsa_circ_0000263 functions as a competitive endogenous RNA to regulate the MDM4 regulator of P53 (*MDM4*) expression as a miR-150-5p sponge, suggesting that the regulatory axis of circ_0000263/miR-150-5p/MDM4 could play an essential role in the pathogenesis and development of cervical cancer by influencing *TP53* gene expression.

A recently identified novel circRNA, circSLC26A4, is upregulated in cervical cancer tissues and cells. The high expression of circSLC26A4 is associated with the poor survival of patients [109]. Experiments of knockdown evidenced the role of circSLC26A4 in inhibiting proliferation, invasion, and tumor growth, both in vitro and in vivo. CircSLC26A4 acted as the sponge of miR-1287-5p, targeting the 30 UTR of homeobox A7 (*HOXA7*) mRNA. The biogenesis of circSLC26A4 is promoted by QKI, an RBP that interacts with the QKI response elements (QREs) in solute carrier family 26 member 4 (*SLC26A4*) gene introns.

Circ_0000745 is upregulated in uterine cervical cancer. Higher expression was found in poorly differentiated tumors or vascular/lymphatic invasion. Circ_0000745 acts as a tumor promoter, enhancing the ability of the cells to proliferate, migrate, and invade [110].

Hsa_circ_0018289 has a critical role in cervical carcinogenesis. The expression of hsa_circ_0018289, overexpressed in 94% of cervical cancer biopsies compared to their adjacent non-tumor tissues, suggests a functional role of circRNAs in cervical cancer tumorigenesis [111]. In vitro and in vivo experiments have shown that hsa_circ_0018289 knockdown inhibits proliferation and suppresses the migration and invasion of cervical tumor cells, suggesting the suppressive role of hsa_circ_0018289 knockdowns on cervical cancer cell aggression. These results provide a new view of circRNAs for cervical cancer carcinogenesis [111]. 

Circulating circRNAs could be a possible tool for diagnosing cervical cancer. Hsa_circ_0101996 and hsa_circ_0101119 could distinguish squamous cervical carcinoma patients from healthy controls, suggesting that the combination of hsa_circ_0101996 and hsa_circ_0101119 can serve as potential biomarkers for squamous cervical carcinoma detection [112].

The role of cSMARCA5 has been evaluated in human cervical cancer [113]. cSMARCA5 is upregulated in cervical cancer tissues and cell lines. The downregulation of cSMARCA5 has a role in transduced cell proliferation rates. miR-432 is the predicted target of cSMARCA5 and can interact with epidermal growth factor receptor (EGFR), influencing its expression. cSMARCA5 induces cervical cancer progression by targeting miR-432 and upregulating the extracellular signal-regulated kinase 1/2 (ERK1/2) signaling pathway.

Hsa_circ_101996 is highly expressed in cervical cancer and associated positively with stage and negatively with survival. This high expression promotes the proliferation, migration, and invasion of cervical cancer. Hsa_circ_101996 binds to miR-8075 like a sponge, inhibiting the regulatory effect on TPX2 microtubule nucleation factor (*TPX2*), resulting in upregulation of TPX2 and tumor progression [114]. Hsa_circRNA_101996, miR-8075, and TPX2 form a network that regulates cervical cancer progression [114].

Hsa_circ_0023404, upregulated in cervical carcinoma tissues, is associated with poor prognosis [115]. The knockdown of hsa_circ_0023404 inhibits the proliferation, migration, and invasion of cervical carcinoma cells. It has been shown that hsa_circ_0023404 promotes transcription factor CP2 (*TFCP2*) expression by inhibiting miR-136 and activating the yes-associated protein (YAP) signaling pathway. The hsa_circ_0023404/miR-136/TFCP2/YAP mechanism appears to be involved in cervical cancer progression, suggesting that this signaling pathway could be a potential target for cervical cancer therapy.

### 8.2. circRNAs and Vulvar, Anal, Vaginal and Penile Cancers

Vulvar cancer is a malignant tumor that accounts for about four percent of female genital cancers [116]. In 2018, there were 44,235 new cases of vulva tumors (0.2% of all cancers) and 15,222 deaths (0.2% of all cancers) [99]. More than 90% of vulva tumors are squamous cell carcinomas (VSCC) [117], with various subtypes: keratinizing, basaloid, or verrucous. A total of 25% of vulvar tumors are associated with HPV infection, and HPV16 and HPV18 contribute to 70% of these [118]. Recent publications have highlighted that HPV in VSCC is associated with a favorable outcome [119,120]. The diagnosis of vulva cancer is carried out through physical examination and detection of HPV in biopsies. The therapy adopted depends on the disease stage, and involves the surgical treatment, radiation therapy, and chemotherapy. circRNAs studies in gynecological tumors have focused mainly on cervical, ovarian, and endometrial cancer [121], but other genital sphere tumors, such as VSCC, also need to be considered, as reported by Huang et al. [40]. So far, there have been no scientific publications regarding the expression of circRNAs in VSCC. Recently, Nakashima et al. studied the significance of the downregulation of circ_0024169 in tissue angiosarcoma [122], postulating that circ_0024169 has a role as a prognostic factor. This paper reports an elevated expression of circ_0024169 in the VSCC cell line, A431, used as a disease control.

Anal cancer is a rare malignant tumor that accounts for about three percent of gastrointestinal tumors [123]. In 2018, there were 48,541 new cases of anal tumors (0.3% of all cancers) and 19,129 deaths (0.2% of all cancers) [99]. Squamous cell carcinoma (SCC) is the predominant tumor type (80% of all cases). There is an association between HPV infection and pre-malignant and malignant lesions of the anus. High-risk human papillomavirus types are responsible for >80% of cases, and HPV16 is the most prevalent type [124]. The prognostic value of HPV in patients with ASCC (anal squamous cell carcinoma) has also been studied in anal cancer. However, the detection of HPV DNA alone is insufficient to classify tumors as HPV-associated; evaluating the marker presence to refine HPV-associated tumor identification is required.

Chamseddin et al. studied the expression of circE7 as a potential biomarker in ASCC [87]. In addition to stabilized markers such as programmed cell death 1 ligand 1 (PD-L1) and glucose transporter type 1 (GLUT1), the expression of circE7, along with the clinical characteristics and OS of patients, were analyzed using a retrospective study on ASCC. High levels of circE7 were predictors of improved OS. An explanation of the phenomenon could be that elevated circE7 levels could allow for greater expression of E7, which could result in a more significant immune response of the host against the tumor. Additionally, the oncoprotein E7 inhibits invasion and metastasis because it inhibits miR-20a [125]. Furthermore, circRNAs, due to their intrinsic functions as miRNA and protein sponges, could have adverse effects on tumor growth [126,127].

Primary vaginal cancer is also a rare tumor that accounts for one to two percent of female genital malignancies. Vaginal squamous cell carcinoma (VaSCC) represents 90% of vaginal cancers [128]. The most common cause of vaginal cancer is HPV infection, and HPV16 was the most prevalent type detected [129]. In 2018, there were 17,600 new cases of vaginal tumors (0.1% of all cancers) and 8062 deaths (0.1% of all cancers) [99]. HPV status has a prognostic significance in VaSCC. HPV-positive early-stage VaSCC has a better prognosis than HPV-negative tumors [130].

Penile squamous cell carcinoma (PSCC) is a rare tumor. In 2018, there were 34,475 new cases of penile tumors (0.2% of all cancers) and 15,138 deaths (0.2% of all cancers) [99]. HPV infection is involved in developing PSCC, and HPV16 is the main type [131]. It is generally diagnosed in the advanced stage, with an unfavorable prognosis. Few biomarkers have been identified and used in clinical practice. Studies on HPV status and survival have produced controversial and unclear results.

Hernandez et al. did not find any relationship between HPV status and survival [132]. Lont et al. instead showed that HPV positivity was linked to improved disease-specific survival [133]. Unfortunately, for these latter types of cancer, there are no data related to the study of circRNAs, either in tissues or in cell lines. Given the importance of the biomarker role that circRNAs can have in neoplastic pathologies, it would be desirable to invest energy into affirming this role in the neoplastic pathologies of the anogenital tract.

## 9. circRNAs and Oropharyngeal and Oral Cancers

More than 830,000 new cases of squamous head and neck cancers (HNSCC), mostly comprising squamous cell carcinoma of the oropharynx (OPSCC) and oral cavity (OSCC), are diagnosed each year in the world with a rising incidence, particularly in developed countries and among young subjects [99]. Oropharyngeal cancers, including SCC of the base of the tongue, soft palate, tonsils, and tonsillar region, accounted for more than 98,000 cases and 48,000 deaths in 2018 [99]. Oral cancers, arising from the lips, buccal mucosa, mobile tongue, and hard palate, accounted for approximately 377,000 cases and 177,000 deaths in 2018 [99]. Smoking habits and alcohol consumption represent the major risk factors for OPSCC and OSCC. Also, HPV infection has been shown to play a major role in the pathogenesis of OPSCC and much less significance in OSCC. Oropharyngeal cancer cases linked to HPV infection are growing. HPVs are now responsible for 30 to 80% of OPSCCs and 6 to 10% of OSCCs in Europe and the USA [134,135,136]. About 60% of HPV-related oropharyngeal cancers are caused by viral genotype 16, and patients have become younger (median age 57 years). HPV-positive patients have a 16 times higher risk of developing oropharyngeal cancer. However, HPV-positive OPSCC has a 58% reduction in the risk of death compared with HPV-negative OPSCC (HR: 0.42, 95%; CI: 0.27–0.66), with a three year overall survival rate (OSR) of 82.4% for HPV-positive disease compared with 57.1% (*p* < 0.001) for HPV-negative disease [137]. Factors including smoking and nodal stage may influence the prognosis in HPV-positive OPSCC [138,139].

Therefore, HPV-positive OPSCC and OSCC represent a specific clinical entity about treatment–response and survival outcome. Mutational profile analysis showed that HPV-associated OPSCC and OSCC have a low number of gene mutations than those associated with other risk factors containing numerous genetic alterations and somatic nucleotide changes, mainly affecting *TP53* and *CDKN2A* tumor suppressor genes [140]. Identifying circRNA signatures in HPV-associated and non-HPV-associated OPSCC and OSCC would be particularly important for patient stratification and appropriate treatments. However, no study has yet investigated their differential expression in virus-associated OPSCC and OSCC and derived cell lines, but several circRNAs have been identified in OSCC, particularly in tongue squamous cell carcinoma (TSCC) and derived cell lines, generally not associated with HPV infection. Nevertheless, the biogenesis, role, and clinical value and function of circRNAs in OSCC remain still unclear [141,142,143,144]. 

TSCC is one of the most frequent tumors of the oral cavity [145]. The five year survival rate is 30 to 50% and is greatly influenced by the high proliferation rate and early lymph-node metastases [146]. Choosing an appropriate treatment for TSCC is a challenge, although significant progress has been made in recent years. The lack of useful targets to control and monitor this pathology represents one of the main problems in treating TSCC [147]. In the study performed by Wei et al., the expression profile of circRNAs in patients with TSCC and adjacent normal tissue was investigated. Circ_081069, the most upregulated, has been extensively studied for its potential role in TSCC. The results obtained suggested that circ_081069 can exert an oncogenic effect by regulating cell proliferation and migration. Circ_081069 interacts with miR-665 as a competing endogenous RNA. The levels of miR-665 were lower in TSCC, and the overexpression of miR-665 caused a decrease in the expression of circ_081069 [148]. No HPV analysis has been performed in such a study.

CircPVT1 was identified as circ6 by Memczak et al. [24], and subsequently, cirPVT1 was named as its host gene [149,150]. Verduci et al. studied the relationship between circPVT1 and mutated *TP53* [151]. Only five tumors tested positive for HPV in their series; therefore, no difference in cirPVT1 expression was evaluated between HPV-positive and HPV-negative cases. The circPVT1 has been expressed in tongue and pharynx cancer cell lines negative for HPV sequences and mutated in the *TP53* gene. By modulating the expression of circPVT1, an increase or decrease in the malignant phenotype of cell lines has been observed. The upregulation of circPVT1 increased the ability of cells to proliferate and form colonies, outlining the role of circPVT1 as an oncogene. CircPVT1 expression is enhanced transcriptionally by the mut-p53/YAP/TEAD complex. Additionally, circPVT1 represses the function of the miR-497-5p as a tumor suppressor, consequently leading to an increase in the expression of genes that regulate cell proliferation, such as aurora kinase A (*AURKA*) and BUB1 mitotic checkpoint serine/threonine kinase (*BUB1*) [152,153].

Su et al. evaluated the expression of has_circ_0055538 in OSCC samples and their biological role in tongue tumor cell lines, negative for HPV. The expression of has_circ_0055538 was significantly lower in tumor tissues than in normal adjacent normal tissues. After overexpression of hsa_circ_0055538 in cells, a decrease in proliferation compared to normal cells, and an increase in apoptosis, was observed. The high expression of hsa_circ_0055538 inhibited the migration and invasion of cells. Hsa_circ_0055538 regulates tumor growth through the p53/Bcl2/caspase signaling pathway, demonstrated by an increase in TP53, BCL2-associated X, apoptosis regulator (BAX), apoptotic peptidase-activating factor 1 (APAF1), caspase 3 (CASP3), and CDKN1A expression, and a decrease in BCL2 expression in the hsa_circ_0055538 overexpressing cells. The opposite results were obtained in the cells in which the expression of hsa_circ_0055538 was knocked down. After overexpressed p53 proliferation, migration and invasion were inhibited in these cells compared to the control cells. Furthermore, the high expression of hsa_circ_0055538 decreased both tumor growth and weight in nude mice [154]. No HPV analysis has been performed in such a study.

Li et al. studied the role of hsa_circ_0008309 in OSCC samples and in tongue tumor cell lines, negative for HPV [155]. Hsa_circ_0008309 was downregulated in cancer tissues. The cells overexpressing hsa_circ_0008309 showed a decrease in the expression of miR-136-5p and miR-382-5p. Bioinformatic analysis showed that Ataxin 1 (*ATXN1*) is strongly associated with these miRNAs. ATXN1, a component of the Notch signaling pathway [156], mediates tumor cell migration and invasion [157]. Additionally, ATXN1 has been shown to regulate proliferation in various cancers [158,159]. Increased ATXN1 protein expression in hsa_circ_0008309 overexpressing cells suggests that hsa_circ_0008309 could regulate the miR-136-5p/miR-382-5p/ATXN1 pathway. Hsa_circ_0008309 acts as a sponge of miR-136-5p and miR-382-5p by modulating the expression of *ATXN1*. No HPV analysis has been performed in such a study.

Circ_0001742 has been reported to be up-regulated in TSCC [160]. Fresh tumor tissues and adjacent non-tumor tissues from patients with TSCC were used to determine the expression of circ_0001742. The expression of circ_0001742 discriminated against the tumor tissues from the adjacent non-tumor tissues. Patients with a higher expression of circ_0001742 were associated with a higher TNM (tumor–node–metastasis) stage. Furthermore, patients with higher circ_0001742 expression had reduced OS compared to those with lower expression. Yao et al. suggested that circ_0001742 may be used as a biomarker for advanced-stage tumors and poor survival. Shao et al. reported that, in TSCC cells, circ_0001742 promotes cell proliferation, invasion, and EMT through targeting the miR-634/RAB1A pathway [161]. Hu et al. identified miR-431-5p as a target of circ_0001742, which regulates the expression of activating transcription factor 3 (*ATF3*). The knockdown of miR-431-5p reversed the effects of circ_0001742 knockdown on TSCC cell proliferation, apoptosis, migration and invasion, and EMT [162]. Additionally, for such a study, no HPV analysis has been performed.

The results of all studies performed in anogenital and oropharyngeal and oral cancers are shown in Table 1.

## 10. Discussion

Despite considerable progress in the classification and biogenesis of circRNAs, efforts still need to be made to clarify the functions of circRNAs better. Due to their multiple functions, circRNAs are involved in many physiological and pathological processes, including viral infections and cancer. In several type of tumors, circRNAs influence the malignant phenotype by regulating specific cancer-related pathways, and exert both an anti-cancer and pro-cancer action by acting as a tumor-suppressor or oncogene [70,71]. Consequently, they can be used as therapeutic targets in treatment. As circRNAs are very resistant to ribonucleases [72], act both in cells and tissues in a specific way, and are stably expressed in biological fluids and exosomes [73,74,75], they could be potential biomarkers in the diagnosis and prognosis of various diseases, including cancers. Recent studies have elucidated the molecular mechanisms responsible for gastric cancer mediated by circRNAs [163,164]. circRNAs are also involved in the pathogenesis of hepatocellular carcinoma (HCC) [165,166]. Differentially expressed circRNAs in HCC hold promise as biomarkers for early diagnosis or prognosis [102,167]. Several circRNAs involved in lung cancer have been identified as non-invasive early biomarkers for diagnosis and prognosis [168]. Dysregulated colorectal cancer-related circRNAs promote tumor progression and/or metastasis [169,170]. Additionally, several circRNAs have been identified that play a role in the tumorigenesis of other malignancies, including bladder and breast cancer, and can be used as biomarkers for diagnosis [171,172].

Worldwide, human papillomaviruses (HPVs) are the most common sexually transmitted infectious agents [173]. Oncogenic HPV genotypes are associated with anogenital and head and neck tumors [174,175,176]. Long-lasting infections with high-risk HPVs can cause tumors in infected organ tissue, such as the cervix, anus, vagina, penis, vulva, and oropharynx [177]. High-risk HPV16 generates circE7, which encompasses the viral E7 oncogene. CircE7 is present in HPV16-transformed cells and is translated to produce E7 protein [178]. CircE7 is required for the optimal expression of E7 in CaSki cells. Consistent with the defined role of E7 in neoplastic transformation, depletion of circE7 causes a decrease in proliferation in these cells. The presence of circE7 in HPV-associated cancers, such as ASCC, is associated with better OS. RNA-seq data present in TCGA from HPV-associated tumors of the head, neck and cervix suggested that elevated levels of circE7 are associated with an improvement in survival [87]. Little is known about the interactions between viruses and host circRNAs; however, it has been reported that in virus-infected cells, the expression patterns of circRNAs are altered compared to control cells [97,98]. The viruses probably use circRNAs for their progression [69,86].

Most circRNAs function as an oncogene in cervical cancer, and are overexpressed in tumor tissues compared to adjacent non-tumor tissues [7,179]. They are involved in tumor progression because they play a role in cell proliferation, migration, and invasion [78]. They almost always perform their functions as sponge miRNAs, regulating specific pathways [180]. circRNAs have often been suggested as biomarkers for diagnosis and prognosis, and as targets of new therapies [76,181]. Although emerging evidence suggests the critical role of circular RNAs (circRNAs) in various malignancies, the biogenesis, role, and clinical value and function of circRNAs in oropharyngeal and oral cancers remain unclear. Some circRNAs are downregulated in oropharyngeal tumor tissues compared to adjacent non-tumor tissues. Their increased expression reduces proliferation, migration, and invasion, and promotes apoptosis. Often, the overexpression of circRNAs in such tumors is associated with poor prognosis.

In HPV-associated tumors, especially in those of the anogenital tract, whose frequency is increasing among the population with a percentage of 0.3% of all tumors, there are no studies on circRNAs as biomarkers. The presence of HPV DNA alone is insufficient to classify tumors as HPV-associated. For these tumors, few markers have been identified and used in clinical practice. Only in anus cancer have circE7 levels been studied as a predictor of improved OS.

## 11. Conclusions

Among the HPV-associated cancers, a fair number of studies concerning the role of circRNAs focused on cervical cancer, and only recently on the oropharyngeal and oral carcinomas. The results obtained so far in cervical tumors have been very promising because they have allowed a clinical application of circRNAs as markers of diagnosis and tumor progression and as probable targets for therapy. There are still few studies relating to the oropharynx carcinomas, but even fewer those relating to vaginal, vulvar, anal, and penile cancers, also because these represent pathologies that have recently been developing in the world. For the latter, very few biomarkers are available, and HPV DNA determination is often used to diagnose. There are also few studies of viral circRNAs, how HPV DNA can modulate the expression of the host circRNAs, and how HPV use them to their advantage. 

Much progress has been made to identify circRNAs, but little is known about the role circRNAs play in tumor development. A better understanding of the mechanisms that regulate the fate of circRNAs and the clinical relevance in tumors will increase the knowledge of the roles that circRNAs have in tumor biology, particularly in HPV-associated tumors, to develop new diagnostic and therapeutic strategies that are much more effective.

## Figures and Tables

**Figure 1 cancers-13-01173-f001:**
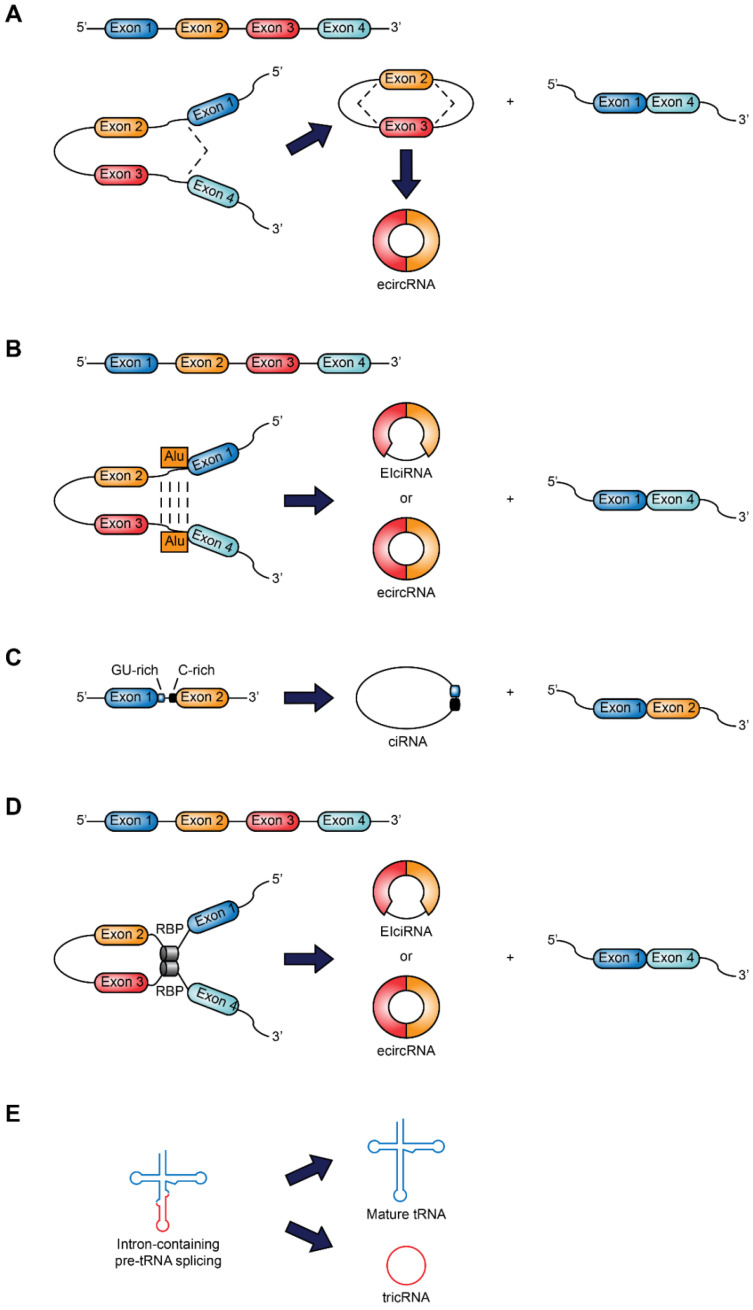
Classification and biogenesis of circRNAs. (**A**) Lariat-guided circularization: an exon skipping event leads to a lariat structure that contains exons 2 and 3, formed by the link between the 3′ end of a donor exon with the 5′ end of the acceptor exon, and a linear product consisting of exons 1 and 4. EcircRNA is formed after removal of introns. (**B**) Circularization is driven by intron-pairing: the pairing of complementary sequences (Alu elements) leads to a circular product and a linear product. Introns are maintained or removed to form an EIciRNA (exon-intron circRNA) or an ecircRNA, respectively. (**C**) Circular intronic RNA (ciRNA): the splicing reaction generates the intron lariat. An element rich in GU, near the junction site 5′, and an item rich in C, near the branching point, make it stable enough to escape from debranching. (**D**) Circularization guided by RNA binding proteins (RBP): the interaction between two RBPs joins two flanking introns to form a circRNA and a linear product. (**E**) tRNA intronic circRNA (tricRNA) originates from pre-tRNA cleaved by the tRNA splicing endonuclease complex.

**Figure 2 cancers-13-01173-f002:**
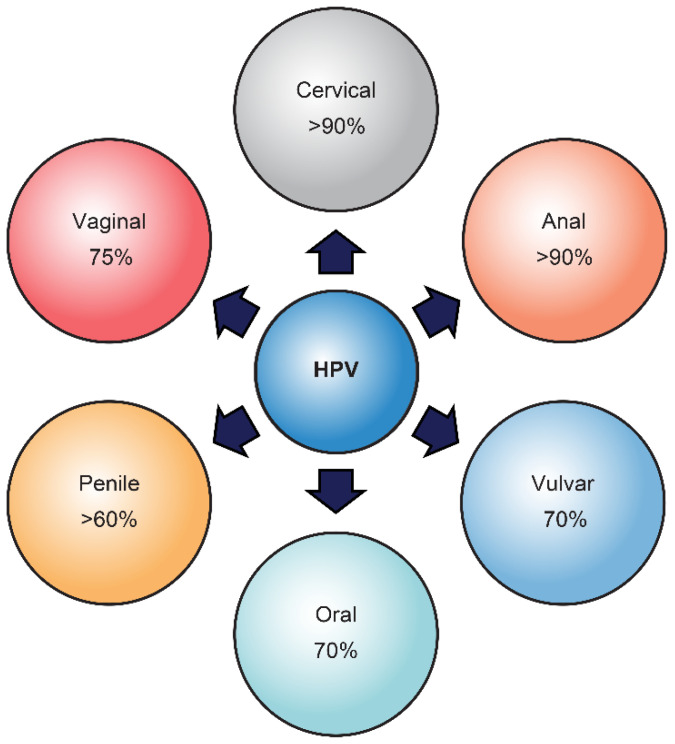
HPV-associated cancers. Types of cancer caused by HPV. The percentage of cancers caused by HPVs is from the United States data from the National Cancer Institute.

**Table 1 cancers-13-01173-t001:** Tumor and/or cell type, expression, functional role, target, and potential use as biomarkers of circRNAs of HPV-associated cancers.

circRNA	Tumor/Cell Type	Expression	Functional Role	Target	Potential Biomarker	Reference
hsa_circ_0001946 (CDR1as/CIRS-7)	Cervical cancers and adjacent normal tissues	Up	Oncogene: promotes proliferation, migration, and invasion	miR-7/*FAK*	Diagnosis, disease progression, and targeted therapy	[104]
hsa_circ8924	Cervical cancers and adjacent normal tissues	Up	Oncogene: promotes migration and invasion	miR-518d-5p/*CBX8*	Diagnosis, disease progression, and targeted therapy	[105]
hsa_circ_000284(circHIPK3)	Cervical cancer and normal cell lines	Up	Oncogene: promotes proliferation and invasion	miR-506/*SNAI2*	Diagnosis, and prognosis	[106,107]
hsa_circ_0000263	Cervical cancer and normal cell lines	Up	Oncogene: promotes proliferation, migration and cell cycle, and inhibits apoptosis	miR-150-5p/*MDM4*	Diagnosis, and treatment	[108]
hsa_circ_0132980 (circSLC26A4)	Cervical cancers and adjacent normal tissues	Up	Oncogene: promotes proliferation and invasion	miR-1287-5p/*HOXA7*	Disease progression, and targeted therapy	[109]
hsa_circ_0000745	Cervical cancers and adjacent normal tissues	Up	Oncogene: promotes proliferation, migration, and invasion	E-cadherin/Vimentin	Prognosis, and targeted therapy	[110]
hsa_circ_0018289	Cervical cancers and adjacent normal tissues	Up	Oncogene: promotes proliferation, migration, and invasion	miR-497	Early disease detection	[111]
hsa_circ_0101996	Cervical cancer and adjacent normal tissues, and peripheral blood of cervical cancer patients and healthy subjects	Up	Oncogene: activates cancer progression	MAPK signaling	Diagnosis	[112]
hsa_circ_0101119	Cervical cancers and adjacent normal tissues, and peripheral blood of cervical cancer patients and healthy subjects	Up	Oncogene: activates cancer progression	MAPK signaling	Diagnosis	[112]
hsa_circ_0001445 (cSMARCA5)	Cervical cancers and adjacent normal tissues, and cervical cancer and normal cell lines	Up	Oncogene: promotes proliferation, migration, and invasion	miR-432/*EGFR*/*ERK1-ERK2*	Disease progression	[113]
hsa_circ_101996	Cervical cancers and adjacent normal tissues	Up	Oncogene: promotes proliferation, migration, and invasion	miR-8075/*TPX2*	Disease progression	[114]
hsa_circ_0023404	Cervical cancers and adjacent normal tissues	Up	Oncogene: promotes proliferation, migration, and invasion	miR-136/*TFCP2*/*YAP*	Targeted therapy	[115]
hsa_circ_0024169	Vaginal carcinoma cell lines	Up	Unknown	Unknown	Unknown	[122]
circE7	Anal cancers and adjacent normal tissues	Up	Unknown	Unknown	Prognosis	[87]
hsa_circ_0055538	Oral cancers and adjacent normal tissues, and tongue cancer cell lines(HPV-negatives)	Down	Tumor suppressor: inhibits proliferation, migration, and invasion, promotes apoptosis	*TP53*/*BCL2*/*CASP3*	Diagnosis, prognosis, and targeted therapy	[154]
hsa_circ_0008309	Oral cancers and adjacent normal tissues, and tongue cancer cell lines (HPV-negatives)	Down	Tumor suppressor: inhibits proliferation, migration, invasion, and EMT	miR-136-5p/miR-382-5p/*ATXN1*	Diagnosis	[155]
hsa_circ_081069	Oral cancer and adjacent normal tissues, and tongue cancer cell lines (HPV-negatives)	Up	Oncogene: promotes proliferation and migration	miR-665	Diagnosis, and targeted therapy	[148]
hsa_circ_0001821 (circPVT1)	Head and neck cancers and adjacent normal tissues, and pharinx and tongue cancer cell lines (HPV-negatives)	Up	Oncogene: promotes proliferation	miR-497-5p/*AURKA*/*BUB1*	Diagnosis, and targeted therapy	[151]
hsa_circ_0001742	Oral cancers and adjacent normal tissues, and tongue cancer cell lines (HPV-negatives)	Up	Oncogene: promotes proliferation, migration, invasion, and EMT	miR-634/*RAB1A*	Prognosis	[160,161]

## Data Availability

Not applicable.

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
