# Peer review of "The Role of circRNAs in Human Papillomavirus (HPV)-Associated Cancers"

_cancers, 2021, doi:10.3390/cancers13051173_

Round 1

Reviewer 1 Report

The paper entitled "The Role of CircRNAs in Human Papillomavirus (HPV)-Associated Cancers", by Bonelli et al., is a detailed review of the works published in the fields of the physiology and of the physiopathology of CircRNAs, especially  in HPV-associated tumors. This review provides an interesting overview of these works, notably through a synthetic table useful for the reader to find data of interest in the different fields of HPV-associated carcinomas.

The quality of the work in its analytical part is unquestionable. There may be a lack of synthesis in the conclusion which is short and difficult to understand (ovarian and endometrial cancers "among the HPV-associated cancers" ?). The paper would be improved with a minimal rewriting of this later part, underlining the major advances aquired in the field as well as the most promising perspectives.

Author Response

Reviewer #1

Reviewer Comment #1. The paper entitled "The Role of CircRNAs in Human Papillomavirus (HPV)-Associated Cancers", by Bonelli et al., is a detailed review of the works published in the fields of the physiology and of the physiopathology of CircRNAs, especially in HPV-associated tumours. This review provides an interesting overview of these works, notably through a synthetic table useful for the reader to find data of interest in the different fields of HPV-associated carcinomas.

The quality of the work in its analytical part is unquestionable.

There may be a lack of synthesis in the conclusion which is short and difficult to understand (ovarian and endometrial cancers "among the HPV-associated cancers"?). The paper would be improved with a minimal rewriting of this later part, underlining the major advances acquired in the field as well as the most promising perspectives.

Author Response #1. We thank the reviewer for the positive comments. We have revised the conclusion and highlighted the significant advances and the most promising perspectives regarding the studies of the role of non-coding RNA in HPV-related cancers.

Reviewer 2 Report

The topic of the manuscript by Bonelli et al. is a very important one, addressing the role of the relatively newly discovered circular RNA (circRNA) species in human papillomaviruses associated cancers. There is in particular a huge interest worldwide on reliable and non-invasive cancer progression markers, especially following the implementation of HPV testing as a primary cervical screening test, for optimal risk assessment and management of patients.

The manuscript of Bonelli et al. is a comprehensive work, but unfortunately, it lacks clarity and focus.

There are many repetitions throughout. The first part describing circRNAs is too long, with many detailed descriptions that are not important for the topic of this review, and that already have been nicely reviewed elsewhere,.

Most importantly, the part that deals with circRNA and HPV-related cancers is de-organized, and partly out of topic, addressing for example potential biomarkers in HPV-associated cancers with no relation to circRNA (vulvar, vaginal and penile cancers), or presenting findings on circRNAs without any indication on whether these findings are for example related to the few HPV-associated oral squamous cell carcinomas.

The descriptions of circRNAs in the different cancer types should have been more structured. These parts are not easy reading, and it is not consistently stated if the findings that are reviewed are from cell culture, tissues, or from peripheral blood/circulating circRNAs. A chapter or section dealing more specifically with the different potential clinical use of circRNAs as therapeutic targets and/or biomarkers could have synthesized the different findings in a more interesting manner.

The aim and topic of the manuscript is therefore not clear, and the authors should decide on what they wish to concentrate on, biomarkers in general, or circRNA in particular, mucosal cancers in general, or HPV associated cancers in particular.

In more details

  1. Generally there are too many irrelevant details in chapter 3, since the primary focus of the review is on HPV-related circRNA. Mentioning functions in general, but not detailing all findings would be more appropriate
  2. Page 6, the section starting with “RBPs play a role in the biogenesis….” And ending with “..human hepatocellular carcinoma” does not deal with the function of circRNA, but with biogenesis and should be removed.  
  3. Last sentence on page 6, and first on page 7 do not relate to the references. The topic is otherwise correctly explained in the review of Kristensen et al., Oncogene volume 37, pages555–565(2018).
  4. Chapter 5: the only reference in this section concerns cancer in general, not only HPV-associated cancers
  5. Chapter 7: The first part of the chapter deals directly with viral circRNA, while the second part deals with alteration of cellular circRNA, related to E7 expression. Even if circE7 seems to be transcribed to functional E7 protein in Caski cells, the second part does not directly relate to circE7, and the heading of the chapter should thus be modified. In addition, the mention to Caski as HPV16 positive cervical cancer cells should come in the first section of this chapter, when Caski cells are mentioned for the first time. The description of the findings of Chamseddin et al. is otherwise repeated in chapter 8.2, page 11.  
  6. The sentence: “the viruses probably use circRNAs for their progression” at the end of this chapter is not clear. Does it relate to cancer progression? Virus replication?
  7. Chapter 8.1. The listing of all circRNAs identified to be modified in cancerous tissues could be structured in a better way. In particular, some findings are from circulating circRNAs, for example the study of Zheng et al on circHIPK3 (reference 111, that is wrongly attributed to Ma et al, together with reference 112). This chapter represents the core information of the review, and the information is nicely summarized in table 1, but this chapter could have been further extended, and the circRNAs involved could have been sorted according to their biomarker potential.
  8. Chapter 8.2. Most of this chapter does not address the role of circRNAs in the HPV associated vulva, vaginal or penile cancer, and most of this chapter is therefore not a part of the main topic of the review, and adds only confusion to the manuscript. The only study on HPV associated anal cancer (Chamseddin et al) is already dealt with in chapter 7.
  9. Chapter 9.1: Only a minor proportion of this cancer type (OSCC) is HPV-associated, and there is no mention of whether the circRNAs identified are related to HPV associated cancers or not. If not HPV-associated, this whole chapter is not part of the stated topic of the review and should be removed.
  10. Chapter 9.2: It is not clear whether the studies of Su et al, and Li et al, on tongue tumor cell lines are HPV associated. If the cell lines are not HPV positive, the findings in these sections are not part of the topic of the review.  

Minor points

Abstract:

  1. has been shown to be expressed and to translate the E7 oncoprotein, should be rephrased: …has been shown to be expressed and to serve as transcript for synthesis of the E7 oncoprotein

Introduction:

  1. The first sentence on page 2: .. originate from new alternative regulated splicing…,  should be rephrased: .. originate from newly identified alternative regulated splicing…
  2. Next sentence on page 2: In the past few years, circRNAs have remained mostly unknown due to the limitations of quantitative reverse transcription PCR (RT-qPCR) techniques, which are generally performed with primers that are unable to distinguish linear from circular RNAs due to being designed on a linear genome,  should be rephrased : In the past few years, circRNAs have remained mostly unrecognized due to the limitations of quantitative reverse transcription PCR (RT-qPCR) techniques, which are generally performed with primers that are unable to distinguish linear from circular RNAs due to being designed from linear genome sequences, rather than “outward facing” for evidencing back-spliced RNA species
  3. Page 5: Cancer-associated chromosomal translocations originate fusion circRNAs that contribute to cell transformation by influencing cell viability and resistance to therapy, and promoting tumor in vivo models  should be rephrased: Cancer-associated chromosomal translocations lead to the production of fusion circRNAs that contribute to cell transformation by influencing cell viability and resistance to therapy, and promoting tumor in vivo models
  4. Page 6: Some circRNAs can modify the stability of mRNAs. CDR1 constitutes a stable duplex, with mRNA stabilizing it. been shown to stabilize CDR1 mRNA.   This sentence is not clear and could be rephrased: A circular antisense RNA molecule has been shown to stabilize CDR1 mRNA
  5. Page 6: Circ-ZNF609 has, in its structure, start and stop codons, for which it can bind more ribosomes. This sentence is not clear and could be rephrased: Circ-ZNF609 contains both a start and a stop codon, and an internal ribosome entry site in its untranslated region, allowing for translation in a cap-independent manner.
  6. Chapter 4: CircRNAs are also insensitive to the action of ribonucleases [74], and their levels of expression are lower than those of mRNAs [27,54,65]. However, circRNAs act in cells and tissues in a type-specific manner, and are stably expressed in saliva, blood, tissues, and exosomes.  The meaning of this section is not Clear

Author Response

Reviewer #2

Reviewer #2 General Comment. The topic of the manuscript by Bonelli et al. is a very important one, addressing the role of the relatively newly discovered circular RNA (circRNA) species in human papillomaviruses associated cancers. There is in particular a huge interest worldwide on reliable and non-invasive cancer progression markers, especially following the implementation of HPV testing as a primary cervical screening test, for optimal risk assessment and management of patients.

The manuscript of Bonelli et al. is a comprehensive work, but unfortunately, it lacks clarity and focus.

There are many repetitions throughout. The first part describing circRNAs is too long, with many detailed descriptions that are not important for the topic of this review, and that already have been nicely reviewed elsewhere.

Most importantly, the part that deals with circRNA and HPV-related cancers is de-organized, and partly out of topic, addressing for example potential biomarkers in HPV-associated cancers with no relation to circRNA (vulvar, vaginal and penile cancers), or presenting findings on circRNAs without any indication on whether these findings are for example related to the few HPV-associated oral squamous cell carcinomas.

The descriptions of circRNAs in the different cancer types should have been more structured. These parts are not easy reading, and it is not consistently stated if the findings that are reviewed are from cell culture, tissues, or from peripheral blood/circulating circRNAs. A chapter or section dealing more specifically with the different potential clinical use of circRNAs as therapeutic targets and/or biomarkers could have synthesized the different findings in a more interesting manner.

The aim and topic of the manuscript is therefore not clear, and the authors should decide on what they wish to concentrate on, biomarkers in general, or circRNA in particular, mucosal cancers in general, or HPV associated cancers in particular.

Author Response. We thank the reviewer for detailed and constructive comments. We have revised the whole text to make more clear and focused the subject of the manuscript. Moreover, we have reduced the text in many parts in order to make the message more fluent.

Reviewer Comment #1. Generally, there are too many irrelevant details in chapter 3, since the primary focus of the review is on HPV-related circRNA. Mentioning functions in general, but not detailing all findings would be more appropriate.

Author Response #1.

The chapter 3 has been significantly reduced by eliminating detailed findings but leaving the general functions.

Reviewer Comment #2. Page 6, the section starting with “RBPs play a role in the biogenesis….” And ending with “..human hepatocellular carcinoma” does not deal with the function of circRNA, but with biogenesis and should be removed.

Author Response #2.

The mentioned section has been modified in order to remove unnecessary information but the function of the RBPs in regulating the circRNAs circularization has been maintained because it is ascribed among the functions.

Reviewer Comment #3. Last sentence on page 6, and first on page 7 do not relate to the references. The topic is otherwise correctly explained in the review of Kristensen et al., Oncogene volume 37, pages555–565(2018).

Author Response #3.

The reference suggested by the reviewer has been added.

Reviewer Comment #4. Chapter 5: the only reference in this section concerns cancer in general, not only HPV-associated cancers

Author Response #4.

In chapter 5, a reference relative to HPV-associated cancers has been added.

Reviewer Comment #5. Chapter 7: The first part of the chapter deals directly with viral circRNA, while the second part deals with alteration of cellular circRNA, related to E7 expression. Even if circE7 seems to be transcribed to functional E7 protein in Caski cells, the second part does not directly relate to circE7, and the heading of the chapter should thus be modified. In addition, the mention to Caski as HPV16 positive cervical cancer cells should come in the first section of this chapter, when Caski cells are mentioned for the first time.

Author Response #5.

The heading of chapter 7 has been modified.

The Caski cells description, as HPV16 positive cervical cancer cells, was moved to the first part of this chapteras suggested by the reviewer.

Reviewer Comment #6. The description of the findings of Chamseddin et al. is otherwise repeated in chapter 8.2, page 11.

Author Response #6.

The repeated text related to Chamseddin et al. has been eliminated.

Reviewer Comment #7. The sentence: “the viruses probably use circRNAs for their progression” at the end of this chapter is not clear. Does it relate to cancer progression? Virus replication?

Author Response #7.

The sentence is related to virus replication and host cell transformation. In this regard, we have added a referencefor an in-depth reading.

Reviewer Comment #8. Chapter 8.1. The listing of all circRNAs identified to be modified in cancerous tissues could be structured in a better way. In particular, some findings are from circulating circRNAs, for example the study of Zheng et al on circHIPK3 (reference 111, that is wrongly attributed to Ma et al, together with reference 112). This chapter represents the core information of the review, and the information is nicely summarized in table 1, but this chapter could have been further extended, and the circRNAs involved could have been sorted according to their biomarker potential.

Author Response #8.

The chapter 8.1 was revised as suggested by the reviewer. The circRNas were sorted in according to their potential biomarker

Reviewer Comment #9. Chapter 8.2. Most of this chapter does not address the role of circRNAs in the HPV associated vulva, vaginal or penile cancer, and most of this chapter is therefore not a part of the main topic of the review, and adds only confusion to the manuscript. The only study on HPV associated anal cancer (Chamseddin et al) is already dealt with in chapter 7.

Author Response #9.

In this section we reviewed available studies on vulvar, anal, vaginal, and penile cancers. For these cancers, few markers are available for diagnosis and prognosis, and we have listed them in the section. While several studies described HPV analysis, very few of them evaluated the presence of circRNAs in these tumors. Therefore, our intention was to highlight the fact that there is not commitment to investigate the role of circRNAs in these tumors, although circRNAS have been shown very promising markers for other tumor pathologies.

Anyway we have shortened this section.

Reviewer Comment #10. Chapter 9.1: Only a minor proportion of this cancer type (OSCC) is HPV-associated, and there is no mention of whether the circRNAs identified are related to HPV associated cancers or not. If not HPV-associated, this whole chapter is not part of the stated topic of the review and should be removed.

Author Response #10.

We agree with the reviewer that being HPV the leading cause of oropharyngeal cancer and being the virus relatively rare in oral cancers, with no studies on circRNAs stratifying by HPV status, the chapter 9.1 need to be removed. We have renamed chapter 9.2 as chapter 9.1 and included few information regarding OSCC at the end of the text related to OPSCC.

Reviewer Comment #11. Chapter 9.2: It is not clear whether the studies of Su et al, and Li et al, on tongue tumour cell lines are HPV associated. If the cell lines are not HPV positive, the findings in these sections are not part of the topic of the review.

Author Response #11.

The cell lines reported by Su et al. and Li et al. are negative for HPV-sequences. However, the information regarding circRNAs studied in these cell lines are relevant to understand differences in HPV-related and HPV-unrelated oropharyngeal carcinoma.

Reviewer Comment #12. Abstract: has been shown to be expressed and to translate the E7 oncoprotein, should be rephrased: …has been shown to be expressed and to serve as transcript for synthesis of the E7 oncoprotein

Author Response #12.

The sentence has been corrected according to the suggestion of the reviewer.

Reviewer Comment #13. Introduction: The first sentence on page 2:  .. originate from new alternative regulated splicing…, should be rephrased:  ..originate from newly identified alternative regulated splicing…

Author Response #13.

The sentence has been corrected according to the suggestion of the reviewer.

Reviewer Comment #14. Next sentence on page 2: In the past few years, circRNAs have remained mostly unknown due to the limitations of quantitative reverse transcription PCR (RT-qPCR) techniques, which are generally performed with primers that are unable to distinguish linear from circular RNAs due to being designed on a linear genome, should be rephrased: In the past few years, circRNAs have remained mostly unrecognized due to the limitations of quantitative reverse transcription PCR (RT-qPCR) techniques, which are generally performed with primers that are unable to distinguish linear from circular RNAs due to being designed from linear genome sequences, rather than “outward facing” for evidencing back-spliced RNA species.

Author Response #14.

The sentence has been corrected according to the suggestion of the reviewer.

Reviewer Comment #15. Page 5: Cancer-associated chromosomal translocations originate fusion circRNAs that contribute to cell transformation by influencing cell viability and resistance to therapy, and promoting tumor in vivo models should be rephrased: Cancer-associated chromosomal translocations lead to the production of fusion circRNAs that contribute to cell transformation by influencing cell viability and resistance to therapy, and promoting tumor in vivo models.

Author Response #15.

The sentence has been corrected according to the suggestion of the reviewer.

Reviewer Comment #16. Page 6: Some circRNAs can modify the stability of mRNAs. CDR1 constitutes a stable duplex, with mRNA stabilizing it. been shown to stabilize CDR1 mRNA. This sentence is not clear and could be rephrased: A circular antisense RNA molecule has been shown to stabilize CDR1 mRNA

Author Response #16.

The sentence has been corrected according to the suggestion of the reviewer.

Reviewer Comment #17. Page 6: Circ-ZNF609 has, in its structure, start and stop codons, for which it can bind more ribosomes. This sentence is not clear and could be rephrased: Circ-ZNF609 contains both a start and a stop codon, and an internal ribosome entry site in its untranslated region, allowing for translation in a cap-independent manner.

Author Response #17.

The sentence has been corrected according to the suggestion of the reviewer.

Reviewer Comment #18. Chapter 4: CircRNAs are also insensitive to the action of ribonucleases [74], and their levels of expression are lower than those of mRNAs [27,54,65]. However, circRNAs act in cells and tissues in a type-specific manner, and are stably expressed in saliva, blood, tissues, and exosomes. The meaning of this section is not Clear

Author Response #18.

The meaning of this sentence is that being the expression of circRNAs specific for tissues/cancer cells, being released in saliva, blood, and in exosomes and generally resistant to ribonucleases, they can be considered excellent biomarker candidates for diagnosis and prognosis of diseases. We have modified the sentence to make clear the meaning.

Reviewer 3 Report

This review provides needed background on circRNAs for those who do not specialize in this field. It also provides information on the interaction of circRNAs with HPV. There are some issues that need to be corrected before it is ready for publication. These issues are mainly concerned with content and editing.

Major Problems

Pages 8 and 9 provide lists circRNAs, but with any further description the list is basically meaningless. Maybe a table or other means be used to provide some idea as to potential functions of these circRNAs.

Section on CircRNAs and Vulvar, Anal, Vaginal, and Penile Cancers needs to be drastically shortened. You state there have been no scientific publications regardingthe expression of circRNAs in VSCC, so this paragraph should be shortened to one sentence to make that statement. The same goes for the paragraphs on vaginal and penile cancers as they also seem to lack any publications on the topic.

Minor Problems:

On page 5 it talks about a “trans-splicing mechanism” where “exons of two different RNAs join”. This should be included in Figure 1.

The first on page 7 states, “Furthermore, tissue circRNAs are so limited that they could not regulate RNA transcription.” This statement needs some explanation, because what is says  would draw into question the significance of a review on the topic if true.

Define “better OS”, may have missed it but could not find where OS was spelled out.

Author Response

Reviewer #3

This review provides needed background on circRNAs for those who do not specialize in this field. It also provides information on the interaction of circRNAs with HPV. There are some issues that need to be corrected before it is ready for publication. These issues are mainly concerned with content and editing.

Author Response. We thank the reviewer for positive and constructive comments. We have revised the whole text in order to address specific issues indicated by reviewer.

Reviewer Comment #1. Pages 8 and 9 provide lists circRNAs, but with any further description the list is basically meaningless. Maybe a table or other means be used to provide some idea as to potential functions of these circRNAs.

Author Response #1.

The list of circRNAs and target miRNAs was deleted, and the sentence was rephrased.

Reviewer Comment #2. Section on CircRNAs and Vulvar, Anal, Vaginal, and Penile Cancers needs to be drastically shortened. You state there have been no scientific publications regarding the expression of circRNAs in VSCC, so this paragraph should be shortened to one sentence to make that statement. The same goes for the paragraphs on vaginal and penile cancers as they also seem to lack any publications on the topic.

Author Response #2

In this section we reviewed available studies on vulvar, anal, vaginal, and penile cancers. For these cancers, few markers are available for diagnosis and prognosis, and we have listed them in the section. While several studies described HPV analysis, very few of them evaluated the presence of circRNAs in these tumors. Therefore, our intention was to highlight the fact that there is not commitment to investigate the role of circRNAs in these tumors, although circRNAS have been shown very promising markers for other tumor pathologies.

Anyway we have shortened this section.

Reviewer Comment #3. On page 5 it talks about a “trans-splicing mechanism” where “exons of two different RNAs join”. This should be included in Figure 1.

Author Response #3

The trans-splicing mechanism was not included in the figure because it is a relatively rare event observed in some eukaryotes (e.g., C. Elegans).

Reviewer Comment #4. The first on page 7 states, “Furthermore, tissue circRNAs are so limited that they could not regulate RNA transcription.” This statement needs some explanation, because what is says would draw into question the significance of a review on the topic if true.

Author Response #4

miRNAs are themselves sequestered and neutralized by their targets, including circRNAs. Since the ability of miRNAs to repress their targets (mRNAs) decreases with the number and abundance of circRNAs, if the amount of circRNAs is low, it follows that transcription regulation is limited. This event is one of the consequences of miRNA-target interactions.

Reviewer Comment #5. Define “better OS”, may have missed it but could not find where OS was spelled out.

Author Response #5

OS is defined in chapter 7, “HPVs and Viral circRNAs,” in the sentence “Patients with elevated circE7 levels had better overall survival (OS), which was related to lower PDL1 expression and a better tumor stage”.

Round 2

Reviewer 2 Report

Bonelli et al. provided a nice revised version of their manuscript. There are however still some minor points that need to be addressed.

First of all, better mention should be made in paragraph 9.1, about whether the findings that are presented in this review actually are from HPV confirmed cases/cell lines or not, as this will imply on the importance of biomarker value of the circRNAs.. In particular, the circRNAs from cell culture studies, by Su et al, and Li et al, are not HPV associated, and these cell cultures actually are models for oral squamous cell carcinomas, rather than oropharyngeal squamous cell carcinomas. The relevance should be better discussed/highlighted in the paragraph, as well as in table 1.

More generally, the carcinogenic processes may differ between HPV-related and non HPV-related OPSCC (and the few HPV related OSCC), and some circRNAs identified may therefore  hold as useful diagnostics or prognostics markers  for HPV-related cancers only, or vice versa. Therefore, mention on whether the studies reviewed in this manuscript have addressed specifically HPV positivity, or whether they do concern any type of OPSCC, or OSCC regardless of HPV status, should be specified.

In addition there are some more details to address:

Page 2, top sentence. The word “identified” is lacking. … originate from newly identified alternative…

Page 2, next sentence: should be “for evidence of”, or “evidencing” in the sentence: “than “outward facing” for evidencing back-spliced RNA species”

Page 13, The last sentence: “The results of all studies performed in anogenital and head and neck cancers are shown in Table 1”, does not belong specifically to paragraph 9, but applies more generally for paragraphs 7, 8 and 9, and this sentence should be moved for example to paragraph 5.

Page 16, Discussion: Several sentences are directly copy/paste from previous parts of the manuscript, and thus they do not add anymore information or insight to the review. These sentences should be either more elaborated in order to shed better light on the topic of the review, or they should be removed.

For example:

  1. the 3 sentences “.. they can be used as therapeutic agents. CircRNAs are insensitive to the action……. tissues, and exosomes”.
  2. "Little is known about…..to control cells”. The sentence that follows: “The viruses probably use circRNAs for their pathogenicity” is, by the way, better formulated than “The viruses probably use circRNAs for their progression [69,86].” as written at the end of paragraph 7

Author Response

Reviewer Comment. First of all, better mention should be made in paragraph 9.1, about whether the findings that are presented in this review actually are from HPV confirmed cases/cell lines or not, as this will imply on the importance of biomarker value of the circRNAs. In particular, the circRNAs from cell culture studies, by Su et al, and Li et al, are not HPV associated, and these cell cultures actually are models for oral squamous cell carcinomas, rather than oropharyngeal squamous cell carcinomas. The relevance should be better discussed/highlighted in the paragraph, as well as in table 1.

More generally, the carcinogenic processes may differ between HPV-related and non HPV-related OPSCC (and the few HPV related OSCC), and some circRNAs identified may therefore hold as useful diagnostics or prognostics markers for HPV-related cancers only, or vice versa. Therefore, mention on whether the studies reviewed in this manuscript have addressed specifically HPV positivity, or whether they do concern any type of OPSCC, or OSCC regardless of HPV status, should be specified.

Author Answer. We agree with the reviewer that biological and molecular differences between HPV-positive and HPV-negative oropharyngeal and oral tumors are relevant considering that they are two clinical and molecular entities. Therefore, the identification of circRNA specific signatures differentially expressed in the HPV-associated and non-HPV-associated tumors would be crucial. According to the reviewer's suggestion we have re-organized the 9.1 paragraph and changed information in Table 1.

All other details were changed as suggested by the reviewer.

Page 2 top sentence: the word "identified" has been added

Page 2 next sentence: the word "evidencing" has been added

Page 13: the sentence has not been moved to paragraph 5 because it is a summary of the results of the studies carried out in all the HPV-associated cancers examined.

Page 16 in Discussion: the sentences have been rephrased